# Avoidance Behavior in Patients with Chronic Dizziness: A Prospective Observational Study

**DOI:** 10.3390/jcm11247473

**Published:** 2022-12-16

**Authors:** Tino Prell, Hubertus Axer

**Affiliations:** 1Department of Geriatrics, Halle University Hospital, 06120 Halle, Germany; 2Center for Healthy Ageing, Jena University Hospital, Friedrich Schiller University, 07747 Jena, Germany; 3Department of Neurology, Jena University Hospital, Friedrich Schiller University, 07747 Jena, Germany

**Keywords:** avoidance behavior, predictors, chronic dizziness, vertigo

## Abstract

Avoidance behavior in adults with chronic dizziness is common. Here, we analyzed factors that are associated with avoidance behavior in a sample of adults with chronic dizziness and/or vertigo. Therefore, 595 patients with chronic vertigo and dizziness who had been subjected to our 5-day multimodal treatment program in a tertiary care outpatient clinic for vertigo and dizziness were prospectively investigated. Both general and dizziness/vertigo-specific data were collected at baseline (*n =* 595) and at 6-month follow-up (*n =* 262). Avoidance behavior was measured using the Mobility Inventory for Agoraphobia (MI). The Hospital Anxiety and Depression Scale was used to estimate anxiety (HADS-A) and depression (HADS-D). At baseline, higher MI (higher level of avoidance) was associated with female gender, higher HADS-D, higher HADS-A, and of a higher age. HADS-D provoked the strongest effect on public places, while gender had the strongest effect on open spaces. The majority (79%) reported improvement of MI and 21% reported that MI had worsened or remained stable at follow-up. In the Generalized Estimating Equations, female gender, higher HADS-A, higher HADS-D, and of a higher age predicted higher MI at follow-up. In particular, older female persons with depressive and anxiety symptoms have a high risk for avoidance behavior related to dizziness.

## 1. Introduction

Vertigo and dizziness are common complaints with a high lifetime prevalence in the general population [1,2]. Dizziness impairs functioning in daily life and is frequently associated with depression, anxiety, and panic [3,4,5,6,7]. People may develop a fear of experiencing dizziness [8,9,10]. As a consequence, people with dizziness often avoid activities that may be associated with the experience of dizziness. Avoidance of genuinely threatening situations characterizes adaptive fear. However, excessive avoidance in the absence of a real threat loses its adaptive value and impairs quality of life [11]. Such maladaptive avoidance is common with a wide spectrum of mental disorders [12].

Avoidance behavior can be classified into different types, including situational avoidance (avoiding people, places, or things), cognitive avoidance (actively turning the mind away from distressing thoughts or memories), or protective avoidance (doing things that help one to feel safer, e.g., compulsive cleaning) [13]. Agoraphobia describes the anxiety that occurs when one is in a situation or setting from which instant escape would be difficult. Individuals with agoraphobia, therefore, aim to avoid such situations or locations [14]. In a study of a self-help group for dizziness, 20.4% of participants were diagnosed with panic syndrome, with or without agoraphobia; and an additional 8.7% were diagnosed with agoraphobia only, totaling nearly 30% with anxiety disorders [15].

Studies addressing avoidance behavior in chronic dizziness are rare. A longitudinal observational study showed that fear-avoidance beliefs in people with vestibular disorders (measured with the Vestibular Activities Avoidance Instrument) predicted limitations in daily activities at 3 months follow-up [3,16]. Another study explored the nature of vestibular-related symptoms in veterans with and without post-traumatic stress disorder. Veterans with increased avoidance reported more vertigo and dizziness-related handicaps than those with post-traumatic stress disorder and reduced avoidance [17].

Although behavioral strategies of avoidance are significant mechanisms for sustaining chronic dizziness, studies on avoidance behavior in chronic dizziness are scarce. Several open questions remain. For example, the influence on avoidance of sociodemographic parameters (age, sex), mental parameters (depression, anxiety), and vertigo/dizziness-specific parameters (classification of vertigo, disease duration, etc.) needs further investigation. This may help us to understand how avoidance behavior can be managed and, finally, how the treatment of patients with chronic dizziness can be improved.

With this explorative study, we focused on the factors associated with avoidance behavior in a sample of adults with chronic dizziness and/or vertigo. In particular, we were interested to see whether avoidance behavior might be associated with dizziness-associated factors (such as attack-like or continuous dizziness, clinical diagnosis) or with gender, age, anxiety, or depression.

## 2. Materials and Methods

This is a secondary analysis of a dataset used in an earlier study of 754 people with chronic dizziness and/or vertigo. The dataset includes baseline data and follow-up data from 6 months after a specialized day-care multimodal treatment program at the Center for Vertigo and Dizziness at Jena University Hospital, Germany [18]. This study was carried out in accordance with the Declaration of Helsinki and all participants gave written informed consent. The study was approved by the local Ethics Committee (number 5426-02/18).

### 2.1. Multimodal Treatment Program

The multimodal and interdisciplinary day-care treatment program for chronic dizziness/vertigo has been already described in detail [18,19]. In short, it was a five-day outpatient program with approximately 7 hours of therapy per day. Physiotherapeutic training, psychoeducation based on cognitive behavioral therapy (CBT) and group therapy, training in Jacobson’s muscle relaxation technique, health education, specialized medical evaluation, and optimized drug therapy were parts of this interdisciplinary therapy program.

### 2.2. Variables

A long-standing and widely used measure of avoidance behavior is the Mobility Inventory for Agoraphobia (MI) [20,21]. The MI can be used to evaluate patients’ avoidance behavior in various situations, both when they are accompanied by someone (Avoidance Accompanied scale) and when they are alone (Avoidance Alone scale), on a scale ranging from 1 (never avoid) to 5 (always avoid) [21].

The MI has excellent psychometric properties [20] and covers a variety of situations (i.e., public places, public transportation, open spaces, enclosed places, cars, etc.). There are heterogeneous results in terms of the underlying factor structure of the MI, ranging from a one-factor model, through a two-factor model (public, crowded, or social situations, and, enclosed or riding situations) [22] and a three-factor model (public places, enclosed spaces, open spaces) [23,24], to a four-factor model (public places, enclosed spaces (including “parking garages”), public transportation, open spaces), with the four-factor model showing the best fit [25]. Several studies indicated that the psychometric properties of the Avoidance Accompanied scale are poorer and, therefore, only the MI Avoidance Alone scale was analyzed in the present study [25].

The MI at baseline was available in 595 subjects from the previously described cohort of 754 people who attended our multimodal treatment program [18]. In 262 subjects, the MI was assessed a second time after 6 months. People who received or did not receive a second MI at follow-up did not differ in terms of vertigo-specific measures (Appendix A (Appendix A)). The difference between MI at baseline and MI at follow-up is termed MI_diff_.

In addition, seven parameters were assessed: age (metric: years); gender (nominal: male or female); duration of vertigo (nominal: entire time interval the patient suffers from dizziness, <6 or >6 months); vertigo diagnosis according to the International Classification of Vestibular Disorders (ICVD) of the Bárány Society; classification of vertigo (nominal: somatic or non-somatic); presence of vertigo attacks (nominal: yes or no); continuity of dizziness (nominal: continuously present, yes or no).

The Hospital Anxiety and Depression Scale (HADS) was used to estimate anxiety and depression [26,27]. The values of the 7 items are summed up scale by scale. This results in two sum values of the sum scales HADS-A (anxiety) and HADS-D (depression) with value ranges of 0–21, whereby higher values indicate greater depressiveness or anxiety. An optimal balance between sensitivity and specificity was achieved where caseness was defined by a score ≥ 8 on both HADS-A and HADS-D [28].

### 2.3. Statistics

All analyses were conducted using IBM SPSS statistics (Version 25; IBM Corporation, Armonk, NY, USA) and JASP (Version 0.16; https://www.jasp-stats.org, accessed on 9 February 2022), JASP team, Amsterdam, The Netherlands). First, descriptive statistics were used to characterize the sample. Normality was tested with the Shapiro–Wilk test. Univariate analyses were performed with *t*-test, U-test, or Spearman correlation, where appropriate. Stepwise linear regression was performed with the AIC selection criterion (Akaike information criterion) to determine predictors of MI and MI_diff_. MANOVA was used to study the effects of several independent variables on the four MI subscales: public places, open spaces, public transportation, and enclosed spaces [25]. Dynamics of MI were studied between the baseline and follow-up using paired Wilcoxon test and Generalized Estimating Equations (GEE) to account for repeated measures and within-person design.

For all analyses, a *p* value < 0.05 was considered statistically significant.

## 3. Results

### 3.1. Factors Associated with Avoidance Behavior at Baseline

Descriptive statistics for the entire cohort are given in Table 1 and for MI items in Appendix A. Using a cut-off score of 1.61 on the MI Avoidance Alone scale [20], 63.9% (*n =* 380) of the subjects had agoraphobia. According to the HADS-A, 56.5% had no anxiety (HADS-A < 7), 21.3% had marginal anxiety (HADS-A 8–10), and 21.3% had anxiety (HADS > 11) (five missing). With regard to depression, 65.2% had no depression (HADS-D < 7), 23% had marginal depression (HADS-D 8–10), and 10.9% had depression (HADS-D > 11).

We first determined the factors that are associated with avoidance behavior (MI) at baseline. In the univariate analyses at baseline, MI correlated moderately with HADS-A (r = 0.341, *p* < 0.001) and HADS-D (r = 0.364, *p* < 0.001), but not with age. MI was higher in female (MI = 2.53, SD = 1.15) compared with male persons (MI = 1.97, SD = 0.97; *p* < *0*.001, rank-biserial correlation = 0.29). However, MI did not differ between somatic vs. non-somatic origin of dizziness (*p* = 0.161), continuous (*p* = 0.319) or attack-like (*p* = 0.746) dizziness, or short vs. long disease duration (*p* = 0.838).

Accordingly, in the linear regression for female gender, higher HADS-D, higher HADS-A, and age were associated with higher avoidance behavior, and accounted for 18% of the MI variance (Table 2).

### 3.2. Factors Associated with the Avoidance of Different Situations

All four MI subscales were highly correlated with one another (Table 3). A one-way MANOVA was used to study how sex, HADS-D, HADS-A, and age differently influenced the four MI subscales. Here, only the HADS-D (F(4, 237) = 8.446, *p* < 0.001, partial η^2^ = 0.125, Wilk’s Λ = 0.875) and gender (F(4, 237) = 6.81, *p* < 0.001, partial η^2^ = 0.103, Wilk’s Λ = 0.897) were associated with the MI subscales. Post-hoc univariate ANOVAs were conducted for every dependent variable. The HADS-D exerted the strongest influence on public places while gender had the strongest influence on open spaces (Table 4).

### 3.3. Predictors of Sustained Avoidance Behavior

The majority (*n =* 207, 79%) reported improvement in MI (MI_diff_ > 0), while 55 (21%) reported that MI worsened or remained stable (MI_diff_ ≤ 0). From baseline to follow-up, the mean MI decreased from 2.317 ± 1.054 to 1.647 ± 0.812 (*p* < 0.001, rank biserial = 0.72) (Figure 1). This change in MI was pronounced in female persons (MI_diff_ = 0.872, SD = 1.045) compared with male persons (MI_diff_ = 0.283, SD = 1.006) (*p* < 0.001, rank biserial = 0.317).

We then determined the baseline factors that predicted improvement in avoidance behavior at follow-up. For this purpose, we performed a GEE on the MI at follow-up using baseline variables. Here, female gender, higher HADS-A, higher HADS-D, and increased age were associated with higher levels of avoidance behavior at follow-up (Table 5).

## 4. Discussion

With this explorative study, we focused on the factors that are associated with avoidance behavior in a sample of adults with chronic dizziness and/or vertigo. In particular, we were interested to see whether avoidance behavior may be associated with different types of chronic dizziness.

Avoidance behavior was common in our cohort of people with chronic dizziness. According to the proposed MI cut-off [20], 63.9% of the studied participants had agoraphobia. Interestingly, avoidance behavior was not associated with the kind of dizziness (attack-like or not) or the duration of dizziness. In the baseline data, depression, anxiety, age, and female gender were the main factors associated with avoidance behavior. Depression was mainly associated with avoidance of public places and female gender with the avoidance of open spaces.

The reciprocal relationship between avoidance and mental health is unsurprising. People with vestibular disorders show increased prevalence of depression, anxiety, and panic when compared with the general population [3,4,5]. Moreover, people with psychiatric disorders, such as anxiety or depression, are at higher risk of having dizziness [6,7].

We did not perform a randomized controlled trial to determine the effect of our specialized multimodal treatment on avoidance behavior. Therefore, we cannot make causal statements if our multimodal treatment decreases the MI from baseline to follow-up. Descriptively we observed an improvement of avoidance behavior at follow-up which in principle fits to the known benefits of a specialized multimodal treatment program [18,29]. However, various degrees of avoidance behavior were still observable after 6 months. In particular, older and female people with higher HADS at baseline were at risk for higher avoidance behavior at follow-up. Given the impact of avoidance behavior on social functioning and quality of life, we therefore should pay special attention to this subgroup of people with chronic dizziness. We need further studies addressing the distinction between adaptive and maladaptive avoidance. Specifically, avoidance coping (disengagement) can be adaptive, works as an effective emotion-focused coping strategy, and may provide a sense of control over the environment and the potential threat [30].

Some limitations of the study must be addressed. Only 44% of the patients filled out the follow-up questionnaires for MI, which leads to selection bias. However, the baseline parameters did not statistically differ between patients with and patients without follow-up data. In addition, the selection of patients subjected to the therapy week may also introduce a type of selection bias, as it is dependent on clinical diagnosis, patient motivation, disability, and the physical and mental independence to participate in an outpatient therapy program.

In addition, the heterogeneity of the diagnoses in the sample taken into consideration (such as benign paroxysmal positional vertigo, central vertigo, multisensory deficit, vestibular neuritis, among other diagnoses) shows the very different possibilities for recovery for each of these diagnoses. In fact, thanks to recent technological advances, it is now possible to refine the vestibular diagnosis and subject the patient to a precise form of early rehabilitation, reducing chronic vertigo as well as avoidance behavior.

## 5. Conclusions

Our data showed that, in particular, older female persons with depressive and anxiety symptoms have a high risk of avoidance behavior relating to dizziness. Measurement of avoidance may provide important prognostic information, suggesting that an assessment of avoidance behavior could be used by clinicians to identify individuals needing tailored and in-depth treatment of chronic dizziness.

## Figures and Tables

**Figure 1 jcm-11-07473-f001:**
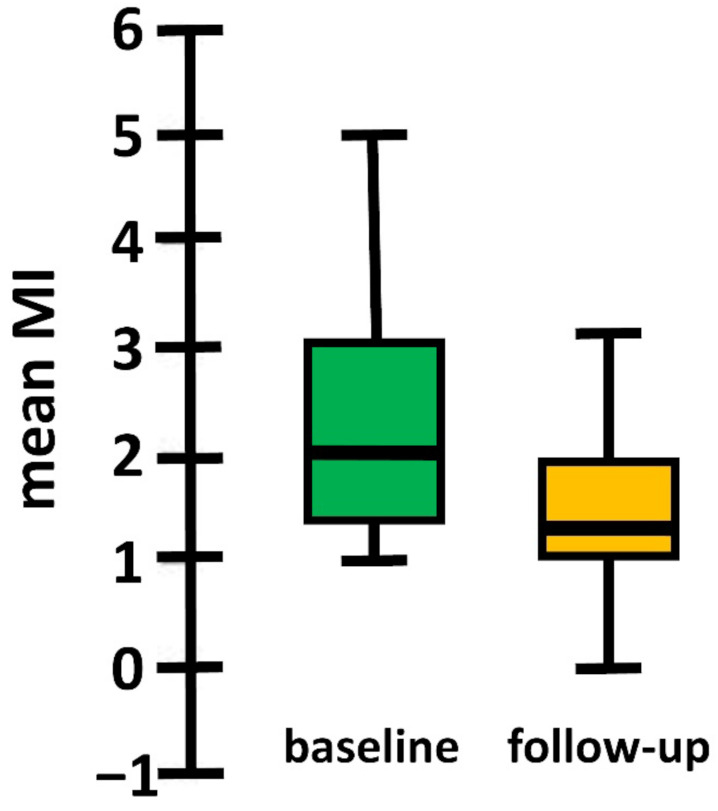
Longitudinal changes of Mobility Inventory for Agoraphobia (MI).

**Table 1 jcm-11-07473-t001:** Descriptive statistics (baseline).

Parameter	Category	*n*	%
Sex	Female	367	62.1
	Male	224	37.9
Clinical diagnosis	BPPV	18	3.0
	BV	26	4.4
	CV	29	4.9
	MD	45	7.6
	MultD	75	12.6
	PPPD	288	48.4
	VM	27	4.5
	VN	65	10.9
	VP	10	1.7
	VS	12	2.0
Somatic vs. non-somatic	Somatic	307	51.6
	Non-somatic	288	48.4
Continuous dizziness/vertigo	Yes	316	55.6
	No	252	44.4
Attacks of dizziness/vertigo	Yes	330	61.9
	No	203	38.1
Duration of dizziness/vertigo	<6 months	79	13.7
	>6 months	496	86.3
		**Mean**	**SD**
HADS anxiety		7.23	4.00
HADS depression		6.34	3.90
Baseline of mean MI alone		2.31	1.12

Abbreviations: BPPV, benign paroxysmal positional vertigo; BV, bilateral vestibulopathy; CV, central vertigo; HADS, Hospital Anxiety and Depression Scale; MD, Meniere’s disease; MI, Mobility Inventor; MultD, multisensory deficit; SD, standard deviation; VM, vestibular migraine; VN, vestibular neuritis; VP, vestibular paroxysmia; VS, vestibular schwannoma.

**Table 2 jcm-11-07473-t002:** Linear regression (MI baseline = dependent variable).

Independent Variable	Coefficient	Beta	*p*
Constant	0.643		0.002
Sex (female)	0.481	0.382	<0.001
HADS-D	0.067	0.303	<0.001
Age	0.011	0.158	<0.001
HADS-A	0.049	0.157	<0.001

Note: Stepwise linear regression. Dependent variable = MI at baseline. Independent variables: age (metric: years), sex (nominal: male or female), duration of vertigo (nominal: <6 or >6 months), classification of vertigo (nominal: somatic or non-somatic), presence of vertigo attacks (nominal: yes or no); continuity of dizziness (nominal: continuously present, yes or no), Hospital Anxiety and Depression Scale (HADS) with its sub-scores for anxiety (HADS-A) and depression (HADS-D). F(4, 590) = 34.715, *p* < 0.001, corrected R^2^ = 0.181.

**Table 3 jcm-11-07473-t003:** Spearman’s correlation coefficients between Mobility Inventory for Agoraphobia subscales (*p* for all < 0.001).

Variable	MI Public Places	MI Open Spaces	MI Open Transportation	MI Enclosed Spaces
MI Public places	--			
MI Open spaces	0.756	--		
MI Open transportation	0.755	0.671	--	
MI Enclosed spaces	0.688	0.711	0.750	--

Abbreviations: MI, Mobility Inventory for Agoraphobia.

**Table 4 jcm-11-07473-t004:** Post-hoc univariate ANOVAs.

	Dependent Variable	F	*p*	Partial eta Squared
Corrected model	MI public places	26.132	<0.001	0.303
	MI open spaces	25.548	<0.001	0.299
	MI open transportation	24.170	<0.001	0.287
	MI enclosed spaces	25.221	<0.001	0.296
Constant	MI public places	14.080	<0.001	0.055
	MI open spaces	30.323	<0.001	0.112
	MI open transportation	12.352	<0.001	0.049
	MI enclosed spaces	5.464	0.020	0.022
HADS-D	MI public places	33.925	<0.001	0.124
	MI open spaces	17.770	<0.001	0.069
	MI open transportation	20.102	<0.001	0.077
	MI enclosed spaces	13.378	<0.001	0.053
Sex	MI public places	8.530	0.004	0.034
	MI open spaces	18.642	<0.001	0.072
	MI open transportation	9.285	0.003	0.037
	MI enclosed spaces	24.018	<0.001	0.091

Abbreviations: HADS-D, Hospital Anxiety and Depression Scale–Depression subscale; MI, Mobility Inventory for Agoraphobia.

**Table 5 jcm-11-07473-t005:** Generalized Estimating Equations on Mobility Inventory for Agoraphobia at Follow-Up.

Parameter	Regression Coefficient	Standard Error	95% Wald Confidence Interval		
			Lower	Upper	Significance
Constant	0.676	0.2710	0.145	1.207	0.013
Sex (female)	0.503	0.0893	0.328	0.678	<0.001
Organic reason for dizziness (yes)	−0.099	0.1005	−0.296	0.098	0.324
Continuous dizziness (yes)	−0.109	0.1226	−0.350	0.131	0.373
Attack-like dizziness (yes)	−0.110	0.1249	−0.355	0.134	0.376
Duration (<6 months)	0.101	0.1379	−0.170	0.371	0.465
Age	0.012	0.0035	0.005	0.019	<0.001
HADS-A	0.053	0.0155	0.023	0.084	<0.001
HADS-D	0.062	0.0153	0.032	0.092	<0.001

Corrected Quasi-likelihood under Independence Model Criterion (QICC) = 505.571. Abbreviations: HADS-D, Hospital Anxiety and Depression Scale–Depression subscale; HADS-A, Hospital Anxiety and Depression Scale–Anxiety subscale.

## Data Availability

The data used to support the findings of this study are available from the corresponding author upon request.

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
