# Peer review of "Avoidance Behavior in Patients with Chronic Dizziness: A Prospective Observational Study"

_jcm, 2022, doi:10.3390/jcm11247473_

Round 1

Reviewer 1 Report

This is a well written article, with a topic of great interest and little studied in the literature. Avoidance behavior is typical in patients with chronic dizziness, but little is known about the factors associated with it. This study highlights these factors, paving the way for a more appropriate treatment for each individual patient with chronic vertigo.

Every single section is clear and exhaustive.

Author Response

Ffirst of all, we would like to thank the reviewers for their very favourable evaluations. In fact, we were inspired to analyse the data regarding avoidance behaviour from our daily clinical experience in the therapy of dizzy patients.

We revised the text to improve English language usage throughout the manuscript.

Reviewer 2 Report

This is a highly interesting paper on a big sample of patients. Topic is important in clinical practice, chronic dizziness impacts the quality of life on many facets. The statistical analysis is thorough, and the results are well documented.

The paper shows convincingly, that the underlying cause of dizziness is far less important for final handicap, then  general psychosocial factors.

Author Response

First of all, we would like to thank the reviewers for their very favourable evaluations. In fact, we were inspired to analyse the data regarding avoidance behaviour from our daily clinical experience in the therapy of dizzy patients. 

We revised the text to improve English language usage throughout the manuscript.

Reviewer 3 Report

Thank you for giving me the opportunity to read and review this manuscript. The topic is relevant for the readers of J of Clin Med but the manuscript has unfortunately flaws. My major concern is that there is no clear aim.

 Specific comments:

Title: Adequate but I suggest adding the design to the title

Abstract: See comments below about aim etc.

Introduction: Gives a good view of the topic.

Aim: There is no aim. The authors give examples of questions that need to be addressed but do not specify any research questions, aims, or hypotheses that this paper will answer.

Method: It is impossible to say if the methods chosen are appropriate for answering the aim since there is no aim. However, variables such as “permanent dizziness” needs to be defined in the method section.

Results: No aim that relates to the results.

Page 4; 3.2: The first sentence is method.

Discussion: The first paragraph in the discussion outlines a possible aim. The first paragraph in the discussion can include the aim but should also summarize the study's main findings.

Page 6, line 182: When refereeing to cross-sectional data, does the authors mean baseline-data? 

Conclusion: The conclusion should clearly answer the questions in the aim, and possibly needs to bee rewritten when the authors has defined their aim.

References: Seems appropriate

Table 1: Permanent dizziness needs to be defined – continuous? Persistent but not the same as duration?

Table 3, 4 and 5: All abbreviations need to be explained in a footnote under all tables.

Figure 1: Three different ways are used to show the same result. Decide which one is the best and use that. Avoid using abbreviations in the heading

Author Response

  1. ‘Title: Adequate but I suggest adding the design to the title’. We added the study design to title (line 2).
  2. ‘Aim: There is no aim. The authors give examples of questions that need to be addressed but do not specify any research questions, aims, or hypotheses that this paper will answer.’. We specified the aim of the study in the abstract (lines 11-12), in the introduction (lines 64-68), in the discussion (line 194-197), and in the conclusion as well (lines 237-238).
  3. ‘.., variables such as “permanent dizziness” needs to be defined in the method section.’ and ‘Table 1: Permanent dizziness needs to be defined – continuous? Persistent but not the same as duration?’ We changed ‘permanent dizziness’ into ‘continuous dizziness’ throughout the manuscript and specified the definitions of duration (lines 106-107) and continuity (line 110) in the methods section.
  4. ‘Page 4; 3.2: The first sentence is method.’ That is correct, we deleted this sentence.
  5. ‘Page 6, line 182: When refereeing to cross-sectional data, does the authors mean baseline-data?’ That is correct we changed ‘cross-sectional’ into ‘baseline data’ (line 201).
  6. ‘Table 3, 4 and 5: All abbreviations need to be explained in a footnote under all tables.’ We changed that accordingly (lines 172, 174-175, 190-192).
  7. ‘Figure 1: Three different ways are used to show the same result. Decide which one is the best and use that.’ We changed figure 1 accordingly.
  8. ‘Avoid using abbreviations in the heading.’ We changed that accordingly (lines 170-171, 188 and 189).

We would like to thank the reviewers for their very favourable evaluations. In fact, we were inspired to analyse the data regarding avoidance behaviour from our daily clinical experience in the therapy of dizzy patients. 

Reviewer 4 Report

Dear Authors and Editors,

Thank you for allowing me to review this interesting manuscript. It follows a previous article published by some of the Authors on this topic. As adequately reported, the psychological and psychiatric problems associated with a balance disorder are very frequent, inducing behavioural strategies of avoidance in the patient. I thank the Authors for both manuscripts, I think it is never enough to underline these aspects related to dizzy patients. As correctly reported, an important limitation of the present study is the fact that the evaluation process was completed only in 44% of cases and this could strongly influence the overall results, as well as the great heterogeneity of the diagnoses in the sample taken into consideration (such as BPPV, CV, MultD, VN..), having each of them an extremely different possibility of recovery. In fact, thanks to recent technological advances, it is now possible to refine the vestibular diagnosis and start the patient to a precise early rehabilitation, reducing chronic vertigo and avoidance behavior and, in my opinion, this must be adequately emphasized in the discussion of this valuable manuscript. The results are presented correctly, the conclusion is focused on the results and the references are up to date.

Author Response

  1. We revised the text to improve English language usage throughout the manuscript.
  1. We specified the meaning of avoidance behaviour in patients with chronic dizziness (in the introduction) to underline its significance (especially lines 29-34 and lines 56-63).
  2. We included the argument in the discussion of the limitations of the study that also ‘the great heterogeneity of the diagnoses in the sample’ has to be discussed, which is very helpful suggestions (see lines 230-235).

We would like to thank the reviewers for their very favourable evaluations. In fact, we were inspired to analyse the data regarding avoidance behaviour from our daily clinical experience in the therapy of dizzy patients.  

Round 2

Reviewer 3 Report

Thank you for our revised manuscript. You still use two ways of showing the same thing in figure 1. I suggest only using boxes and whiskers, which is the most informative way to display the results. If there is a reason for showing both diagrams in the figure, this has to be stated in the title. Otherwise, all my comments have been answered.

Author Response

Yes, you are right. We changed figure 1 accordingly.